# Dendrochronological Analysis of One-Seeded and Intermediate Hawthorn Response to Climate in Poland

Anna Cedro * and Bernard Cedro

Institute of Marine & Environmental Sciences, University of Szczecin, Adama Mickiewicza 16, 70-383 Szczecin, Poland; bernard.cedro@usz.edu.pl
* Correspondence: anna.cedro@usz.edu.pl

**Abstract:** Although the hawthorn is not a forest-forming species, and it has no high economic significance, it is a very valuable component of forests, mid-field woodlots or roadside avenues. The literature, however, lacks information on the growth rate, growth phases, or growth–climate–habitat relationship for trees of this genus. This work aimed to establish the rate of growth of *Craraegus monogyna* and *C. xmedia* Bechst growing in various parts of Poland, in various habitats; analyze the growth–climate relationship; and distinguish dendrochronological regions for these species. Samples were taken using a Pressler borer from nine populations growing in different parts of Poland, from a total of 192 trees (359 samples). The tree-ring width was measured down to 0.01 mm. The average tree-ring width in the studied hawthorn populations ranged from 1.42 to 3.25 mm/year. Using well-established cross-dating methods, nine local chronologies were compiled with tree ages between 45 and 72 years. Dendroclimatic analyses (pointer year analysis, correlation and response function analysis) were performed for a 33-year period from 1988 to 2020, for which all local chronologies displayed EPS > 0.85. The tree-ring width in the hawthorn populations depended mostly on temperature and rainfall through the May–August period. High rainfall and the lack of heat waves through these months cause an increase in cambial activity and the formation of wide tree rings. Conversely, rainfall shortages through this period, in conjunction with high air temperatures, caused growth depressions. Cluster analysis enabled the identification of two dendrochronological regions among the hawthorn in Poland: a western and eastern region, and a single site (CI), whose separation was most likely caused by contrasting habitat and genetic conditions. The obtained results highlight the need for further study of these species in Poland and other countries.

**Keywords:** one-seeded hawthorn (*Crataegus monogyna*); intermediate hawthorn (*Crataegus media* Bechst.); tree-ring width; dendroclimatology; Poland

## 1. Introduction

In recent decades, we have witnessed climate change, manifested mostly via air temperature increase (warmer winters, earlier and longer warm seasons, elevated maximum temperatures), and increasingly extreme weather (e.g., heavy rains, droughts, heat waves) [1]. Also, the human impact on the natural environment has been increasing: occupation, transforming and destroying plant and animal habitats, emissions of substances that are harmful to living organisms into soil, water and air, or introducing organisms into geographic regions where they are not indigenous [2–5]. As a result, the ranges of individual species occurrence are shifting, new threats are occurring in areas where they were previously unknown, and phenomena leading to habitat degradation and species extinction are intensifying [6–9]. Trees and forests are just as vulnerable to these changes as other organisms. At the same time, they represent a very valuable element in counteracting global warming as organisms capable of absorbing and storing carbon dioxide, as organisms/habitats creating a microclimate that is favorable to other organisms, retaining water in the habitat, or lowering the surface temperature [10–12]. The list of ecosystem services



for trees and forest habitats is very long. For this reason, the protection of each species and each forest habitat is so important. The hawthorn, although neither economically significant nor forest forming, is remarkably valuable ecologically. It most commonly inhabits forest edge zones and grows in clearings, mid-field woodlots, or along roads. The hawthorn trees provide habitat for numerous animal species, and offer both nutrition and shelter [13–17]. By increasing the biodiversity of their occurrence sites, they increase the immunity of such habitats to the ongoing changes, and thus increase their quality and value [18]. For these reasons, each new information on the hawthorn ecology is valuable and may be used for counteracting climate change. Two hawthorn species occur in Poland: the one-seeded hawthorn (*Crataegus monogyna*) and the Midland hawthorn (*C. laevigata*). There is also a natural hybrid of these two, the intermediate hawthorn (*C. xmedia* Bechst.). All these species occur in the study area. In the literature, the hawthorn is described as having low requirements with respect to habitat (resistant to both strong frost and drought), growing mostly in sun-lit places [19,20]. The hawthorn rapidly inhabits deforested sites or abandoned agricultural land [21]. It is noted for its high biocenotic significance, as hawthorn gatherings are a source of nutrition for numerous animals, provide shelter, and protect water and soils [22,23]. On the other hand, however, they are also the habitat of numerous pests that infest both hawthorn individuals and orchard-grown trees (especially apple and pear trees). The hawthorn (both inflorescences and fruits) is also used in folk medicine and herbalism as a remedy for diarrhea and insomnia, and for the treatment of cardiovascular diseases and digestive tract conditions. It is the source of numerous highly biologically active chemical compounds that display, for instance, anti-inflammatory, antibacterial and anti-oxidative effects [24–26].

For these reasons, each new information on the hawthorn ecology is valuable and may be used in the fight against climate change. There are few papers on the ecology of the individual hawthorn species, and information regarding tree-ring width, growth rate and growth–climate relationship is virtually absent (apart from Cedro and Cedro [27]).

This paper aims to (i) determine the rate of growth in the hawthorns growing in various parts of Poland, in different habitats; (ii) analyze the growth–climate relationship and (iii) attempt an identification of dendrochronological regionalization for this species.

## 2. Material and Methods

### 2.1. Study Area

The fieldwork focused on nine hawthorn populations in Poland. Two populations (LB and ST) are located in northern Poland (Figure 1), within the young glacial relief zone (the last stagnation phases of the ice sheet). Four populations (MA, CI, ZB and DB) are located within the lowlands of central Poland (70–130 m a.s.l.), and three populations are located within mountain areas (above 450 m a.s.l.): WA in the Sudetes, and WG and LE in the Carpathians. The ST study plot is located at the lowest elevation (45 m a.s.l.), and the LE plot is located at the highest elevation (540 m a.s.l.) (Table 1). Two populations are natural hybrids of the one-seeded hawthorn and the Midland hawthorn, i.e., intermediate hawthorn (*C. xmedia*, ST and LB). The remaining seven plots are populations of the one-seeded hawthorn (*C. monogyna*).

The intermediate hawthorn plot (*C. xmedia*)—LB—is located about 7 km SE from the city of Lębork, in northern Poland, within the Kaszuby Lake District, in a region characterized by extremely variable surface elevation [28]. The studied trees, however, are growing on an area with less variable elevation, on a gentle slope of a small stream (about 120 m a.s.l.), on a Quaternary, mostly sandy substratum. It is the margin of a former military training site. At present, the training site is disused by the military, and it is undergoing a natural succession. The studied trees are growing on clearings, along the forest edge, and underneath the canopy of older trees. The largest group of trees was probably planted, as an espalier arrangement is preserved.

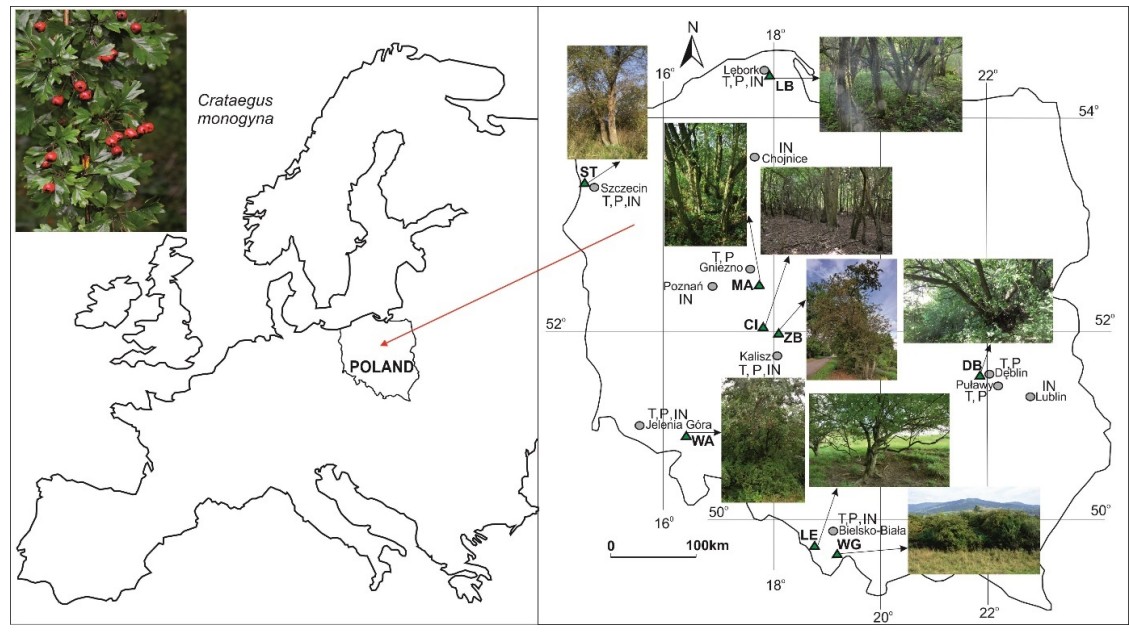

**Figure 1.** Location of the study plots: triangles—study plots, dots—weather stations, T—temperature, P—precipitation, IN—insolation.

**Table 1.** List of the study plots along with basic information.

| Lab. Code | Name | Species | Geographic Coordinates | Altitude a.s.l. (m) | No. of Trees | No. of Samples | No. of Tree Rings |
|---|---|---|---|---|---|---|---|
| LB | Lębork | *C.xmedia* | N: 54.5265° N E: 17.8470° E | 120 | 21 | 41 | 1584 |
| ST | Stobno | *C.xmedia* | N: 53.4181° N E: 14.4051° E | 45 | 22 | 30 | 1990 |
| MA | Malczewo | *C. monogyna* | N: 52.4350° N E: 17.6433° E | 123 | 22 | 40 | 1501 |
| CI | Ciemierów | *C. monogyna* | N: 52.1047° N E: 17.7267° E | 78 | 20 | 38 | 2080 |
| ZB | Zbiersk | *C. monogyna* | N: 51.9724° N E: 18.1124° E | 120 | 21 | 41 | 1508 |
| DB | Dęblin | *C. monogyna* | N: 51.5490° N E: 21.8282° E | 115 | 22 | 42 | 1381 |
| WA | Wałbrzych | *C. monogyna* | N: 50.7995° N E: 16.2333° E | 481 | 21 | 41 | 1327 |
| WG | Węgierska Górka | *C. monogyna* | N: 49.6016° N E: 19.1010° E | 451 | 22 | 44 | 935 |
| LE | Leszna Górna | *C. monogyna* | N: 49.6975° N E: 18.7318° E | 540 | 21 | 42 | 1944 |
| | | | | Σ | 192 | 359 | 14,250 |

The ST study plot is located in the NW part of Poland, on a morainic plateau composed of Quaternary deposits of the last Ice Age [28], in a typically agricultural landscape. On German topographic maps (e.g., from 1921), a road passes through the center of the study plot. Following World War II, the road was plowed and at present it is annually sown. Only the study plot was excluded from farming. The gathering of the intermediate hawthorns is probably derived from one to several individuals sown by birds next to a dirt road. At

present it is a monospecific mid-field woodplot composed of the intermediate hawthorn (*C. xmedia*) trees and shrubs, comprising several hundred individuals of variable age. The woodlot is about 75 m long and 20–25 m wide. It is located on a SE-facing scarp that is up to 4 m high (about 45 m a.s.l.). Cambisols have developed on the till bedrock. The woodlot is surrounded on all sides by agricultural land [27].

The MA study plot, close to Malczewo, is inhabited by the one-seeded hawthorn (*C. monogyna*). It is located in Greater Poland, on the Września Plain (123 m a.s.l.), in a typically agricultural landscape. In between the fields, there is a small pine forest (about 80 years old), with an admixture of ash. A gathering of the hawthorn, the black elderberry and the bird cherry is growing at the edge of this forest, along a dirt road. The forest is growing within a shallow depression, with no streams nearby. The hawthorn occurs mostly at the NE edge of the forest.

The CI study plot is located close to Ciemierów on the South Great Polish Lowland (78 m a.s.l.). The one-seeded hawthorn trees are growing under the canopy of a mixed fresh forest (e.g., with pine, oak and birch) on an abandoned agricultural land. The remains of an old forester's lodge are located nearby. The trees are growing mostly in an avenue layout and were most likely planted by a forester. Gleyed pseudopodzolic soil is developed on clayey sands at this plot.

Also, the ZB study plot, close to Zbiersk, is located on the South Great Polish Lowland (120 m a.s.l.). The one-seeded hawthorns were planted here by foresters in the 1960s–1970s. At present, the trees are growing at the border of a lumber mill and gardens, forming a ~100 m long espalier.

The one-seeded hawthorn trees (*C. monogyna*) from the DB study plot are growing on a flood embankment along the Wisła River (left bank of the river, about 100–300 m to the south of a railroad crossing close to Dęblin, 115 m a.s.l.). Geographically, this area is considered part of the Middle Vistula Valley [28]. A very dense assemblage of the hawthorn is growing at the summit and the slopes of the scarp.

The WA site, located within the city limits of Wałbrzych, is located in the Wałbrzych Mountains, part of the Central Sudetes [28]. In the past, this area hosted numerous pollution-emitting industrial plants, including mines and coking plants. The one-seeded hawthorn trees (*C. monogyna*) are growing next to a disused railway leading to Biały Kamień. The hawthorn individuals occur on gentle slopes and scarps (481 m a.s.l.). The bedrock is composed of Carboniferous-aged sandstones. The trees are growing in clearings, in small woodlot patches, or at forest edges. The population is dominated by young and very young individuals (up to 20–30 years old), only a few specimens are older.

The WG study plot, close to Węgierska Górka (451 m a.s.l.), is located in the Żywiec Basin, part of the Western Beskidy range within the Carpathians. A this site, a dynamically developing population of the one-seeded hawthorn (numerous trees and shrubs, mostly young, with few older specimens) is growing on gentle, south-facing, meadowy slopes. Numerous bunkers, part of the Polish defensive system from the 1930s, are located immediately adjacent to the hawthorn population, and during the sampling campaign for this study (August 2021), intense express road construction was in progress in the vicinity, involving a tunnel, a viaduct and the road. The bedrock is composed of marly shales of *Cretaceous* age.

The one-seeded hawthorn from the vicinity of Leszna Górna (LE study plot) is growing on gentle slopes and near the summits of the elevations forming part of the Silesian Beskid Mountains (part of Western Beskidy, Carpathians, 508–540 m a.s.l.). The woodlots are surrounded by pastures used for sheep farming, and the sheep tend to rest underneath the hawthorn trees. Under the oldest hawthorn specimens, there is often no vegetation under the canopy, the root zones are exposed, and there are well-trodden paths. All this indicates that numerous generations of sheep took rest there. Upper *Cretaceous* shales make up the bedrock at this site.

During sampling in northern Poland (LB and ST study plots), we observed numerous individuals of the hawthorn trees infested by the orchard ermine (*Yponomeuta padella*). This insect causes partial damage to the assimilation apparatus [29,30].

## 2.2. Tree-Ring Data

Samples were taken in September 2020 (the paper by Cedro and Cedro [27] presents residual RES chronology and tree-ring/climate analysis for the period of 1981–2020, 40 years, for the ST site), during the growing season 2021 (MA, CI, ZB, DB, WG and LE), and in late September 2021 (LB and WA). Hawthorn individuals who were dominant and presumably older were selected for sampling. Samples were collected using a Pressler borer, at 1.0–1.3 m height above ground (two cores per individual). A total of 192 trees were sampled (from 20 to 22 per plot), yielding 359 measuring radii (from 30 to 44 per plot). In the laboratory, samples were glued onto boards, dried and sliced with a knife in order to obtain a clear view of the tree rings. In the case of the hawthorn, due to the very weak visibility of annual growth rings, the measurements were carried out using an aqueous filter, and/or the sample surfaces were smeared with chalk. The tree-ring width was measured under a stereomicroscope with an accuracy of 0.01 mm using LDB_Measure software [31]. A total of 14,250 rings were measured. As a next step, local chronologies were compiled using well-established cross-dating methods. Based on the high visual similarity of dendrochronological curves and high values of statistical indicators (Student's *t*-test and correlation coefficient), dendrochronological sequences were selected for building the chronologies. The least visually and statistically correlated sequences were discarded. The quality of the chronologies was tested using COFECHA, part of the DPL software package [32–35]. Student's *t*-test and coherence coefficient (*Gleichläufigkeitswert*, GL) were computed for pairs of chronologies using the TCS 1.0 program [36], in order to determine the similarity between local chronologies. The EPS coefficient was also computed [37]. Age trend and autocorrelation were subsequently removed from the dendrochronological sequences selected for the chronology using an indexing process (a two-phase detrending technique, by fitting either a modified negative exponential curve or a regression line with a negative or zero slope) [33,35]. Standardized (STD) chronologies were selected for dendroclimatic analyses. The period of 1988–2020 (33 years) was adopted for common analyses of chronological similarity and dendroclimatological analyses: pointer years and correlation and response function analyses. Average monthly air temperatures, monthly rainfall totals and monthly insolation values from June of the year preceding growth (pVI) to September of the growing year (IX) were used to analyze correlations and response functions. The analysis was carried out separately for temperature, precipitation and insolation, resulting in $r^2$ values (coefficient of determination of multiple regression) for each meteorological parameter. The analysis of pointer years was carried out using the TCS program [36]. This was achieved by calculating positive years (+) characterized by an increase in the width of the rings in relation to the previous year and negative years (−) in which the rings decreased in width compared to the previous year [38,39]. Pointer years were calculated based on a minimum of 10 trees, using 90% as the minimum threshold for consistency of the growth trend. As regional pointer years, we considered years in which at least 6 local chronologies (out of 9) had a consistent pointer year in the entire study area.

## 2.3. Climate Data

Mean monthly air temperature data (T), monthly precipitation sums (P) and monthly insolation data (IN) were retrieved from 16 weather stations of the Institute of Meteorology and Water Management (IMGW). These originated from stations Lębork—12125 (T, P, IN); Chojnice—12235 (IN); Szczecin—12205 (T, P, IN); Gniezno—252170110 (T, P); Poznań—12330 (T, P, IN); Kalisz—12435 (T, P, IN); Dęblin—12490 (T, P); Puławy—12491 (T, P); Lublin—12495 (IN); Jelenia Góra—12500 (T, P, IN); and Bielsko-Biała—12600 (T, P, IN) (Figure 1), located as near as possible to the respective study plots (from 3.5 to 57 km).

## 3. Results

### 3.1. Ring Width Chronologies

A local chronology was compiled for each study site (Table 2). The longest chronology was obtained for LE (72 years from 1949 to 2020), and the shortest chronology was obtained for MA (45 years from 1976 to 2020). The average age of the studied hawthorn trees equals just 58 years; the studied populations are therefore young and dynamically developing assemblages of this species. The number of samples included in individual local chronologies varies from 13 (for LE) to 24 (for ST), an average of 17 samples. The average tree ring width is the lowest for the WA population (1.42 mm/year), and the highest for the WG population (3.25 mm/year). The average tree-ring width for all the studied populations equals 2.05 mm/year. The rate of tree growth is well represented by cumulative radial growth: the WG population displays the highest growth rate throughout the entire study period, and the WA population displays the lowest growth rate throughout the study period (Figure 2). EPS values > 0.85 are noted from 1972 to 2020/2021. However, the period with EPS > 0.85 is the shortest for the DB site (1988–2020, 33 years). For this reason, this period was assumed as the time frame for the analyses of chronology convergence and the dendroclimatic analyses: pointer year analysis and correlation and response function analysis, which were performed for all study plots.

The chronology convergence was analyzed using the t coefficient and GL [40,41]. The highest t value was obtained for the DB and MA chronologies (10.99), and t > 9.0 was obtained for the following pairs of chronologies: CI and LB, WA and WG, and LE and WA. The lowest t values were obtained for the CI and WA chronologies (2.16), and the pairs of chronologies: LB and WG, ST and WG, and CI and WA are characterized by t values < 3.0 (Table 3). The highest GL value was noted for the same pair of chronologies as in the case of the t index: DB and MA (96%). GL > 80% values were obtained for the following pairs of chronologies: CI and LB, MA and ST, MA and WA, and LE and ST. The GL value is the lowest for the following pairs of chronologies: DB and LB, CI and DB, CI and ST, and CI and MA (66%). Comparably low GL values (GL < 70%) were noted for LB and WA, CI and WG, and CI and LE (Table 3).

**Table 2.** Basic statistics of measured and index (standard) hawthorn local chronologies. Abbreviations: TRW—tree-ring width; SD—standard deviation; 1AC—first-order autocorrelation; MS—mean sensitivity; EPS—Expressed Population Signal.

| Lab. Code | No. of Years | Time Span | No. of Samples | Mean TRW (Min−Max) (mm) | Measured Chronology | | | Standard Chronology | | | EPS >0.85 |
|---|---|---|---|---|---|---|---|---|---|---|---|
| | | | | | SD | 1AC | MS | SD | 1AC | MS | |
| LB | 63 | 1959−2021 | 15 | 1.79 (0.95−3.39) | 0.902 | 0.571 | 0.363 | 0.226 | −0.062 | 0.294 | 1986–2021 |
| ST | 56 | 1965−2020 | 24 | 2.41 (1.48−4.44) | 1.587 | 0.509 | 0.453 | 0.320 | 0.224 | 0.355 | 1981–2020 |
| MA | 45 | 1976–2020 | 16 | 2.38 (1.37−3.84) | 1.491 | 0.531 | 0.440 | 0.290 | 0.088 | 0.334 | 1987–2020 |
| CI | 70 | 1951−2020 | 15 | 1.58 (1.11−2.34) | 0.967 | 0.494 | 0.439 | 0.306 | 0.142 | 0.338 | 1972–2020 |
| ZB | 47 | 1974−2020 | 19 | 1.83 (1.12−3.00) | 1.426 | 0.580 | 0.509 | 0.332 | 0.070 | 0.411 | 1985–2020 |
| DB | 56 | 1965–2020 | 17 | 2.34 (1.02−4.51) | 1.392 | 0.318 | 0.507 | 0.320 | −0.017 | 0.391 | 1988–2020 |
| WA | 62 | 1960−2021 | 18 | 1.42 (0.68−4.30) | 0.906 | 0.435 | 0.491 | 0.332 | 0.007 | 0.429 | 1985–2021 |
| WG | 50 | 1971−2020 | 16 | 3.25 (1.66−4.19) | 1.581 | 0.556 | 0.381 | 0.320 | 0.368 | 0.288 | 1986–2020 |
| LE | 72 | 1949−2020 | 13 | 1.46 (0.76−2.39) | 0.902 | 0.571 | 0.363 | 0.226 | −0.062 | 0.294 | 1976–2020 |

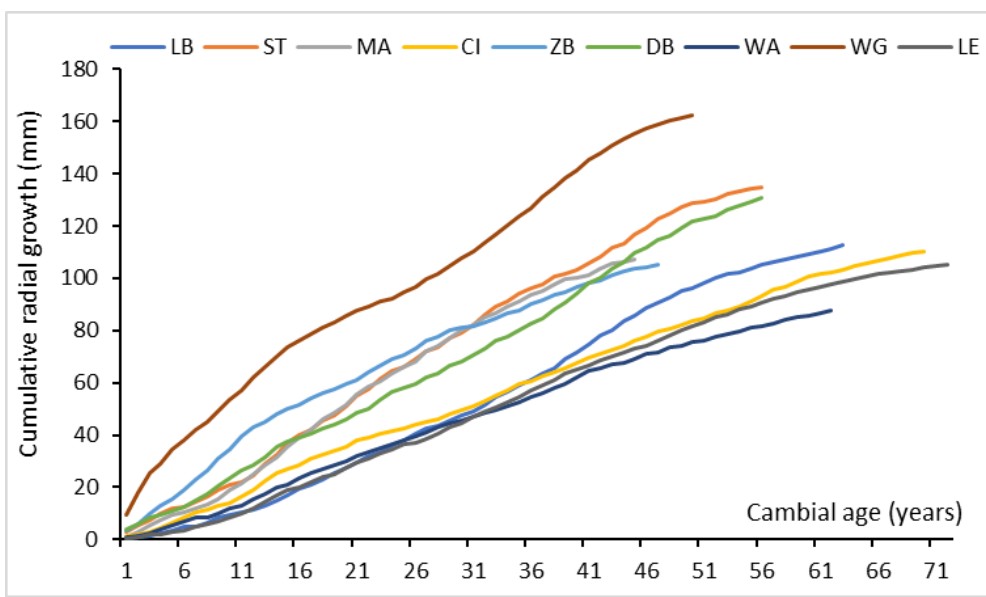

**Figure 2.** Cumulative radial growth of the hawthorn in Poland.

**Table 3.** Convergence of local chronologies of hawthorn as measured with *t* and *GL* (%) values.

| t/GL | LB | ST | MA | CI | ZB | DB | WA | WG | LE |
|------|-----|------|------|------|------|-------|------|------|------|
| LB | - | 4.49 | 3.35 | 9.44 | 3.30 | 4.22 | 3.05 | 2.93 | 3.19 |
| ST | 71 | - | 9.07 | 5.31 | 7.32 | 5.13 | 4.34 | 2.57 | 4.82 |
| MA | 71 | 86 | - | 4.45 | 3.06 | 10.99 | 5.28 | 3.63 | 7.57 |
| CI | 82 | 66 | 66 | - | 5.14 | 4.33 | 2.16 | 3.60 | 3.41 |
| ZB | 74 | 80 | 75 | 74 | - | 2.64 | 4.15 | 5.20 | 3.82 |
| DB | 66 | 76 | 96 | 66 | 72 | - | 3.17 | 3.34 | 4.34 |
| WA | 67 | 80 | 82 | 73 | 76 | 80 | - | 9.88 | 9.49 |
| WG | 72 | 74 | 75 | 69 | 74 | 69 | 78 | - | 7.50 |
| LE | 71 | 86 | 80 | 67 | 76 | 71 | 82 | 80 | - |

### 3.2. Correlation and Response Function Analysis

In the correlation and response function analysis, only negative values are noted as statistically significant for air temperature (Figure 3). In the winter period (December of the previous year, pXII, and January, I), there are single negative correlations. From May to August, however, negative values were noted at all study sites. Higher temperature does not favor the formation of wide rings. The average determination coefficient for all populations equals 28% (from 14% for LB to 36% for LE), and this is the lowest value among the analyzed relationships for the three weather elements.

Positive values of correlation coefficients prevail for precipitation. In the summer of the year preceding growth, there are positive (especially in August) and negative correlation values. From May to July, however, positive correlation values are observed for each site. Positive values indicate that tree-ring width increases with precipitation. On average, the $r^2$ coefficient equals 45% (from 34 to 53%), and this is the highest value among the analyzed weather elements.

Finding a clear relationship pattern in the analysis of the relationship between insolation and growth is challenging. Consistent correlations occur only for three months: negative correlation values in February, positive values in April and negative values in June. These, however, occur simultaneously only in 3 out of 9 chronologies. In the remaining months, the values are either positive or negative, and the relationships are site specific. On average, the $r^2$ value equals 35%, ranging from 18 to 58%. The highest determination coefficient (58%) was observed for the CI plot. At this site, there are only three positive

values, for July and December of the preceding year (pVII, pXII), and for April of the current year (IV) (Figure 3).

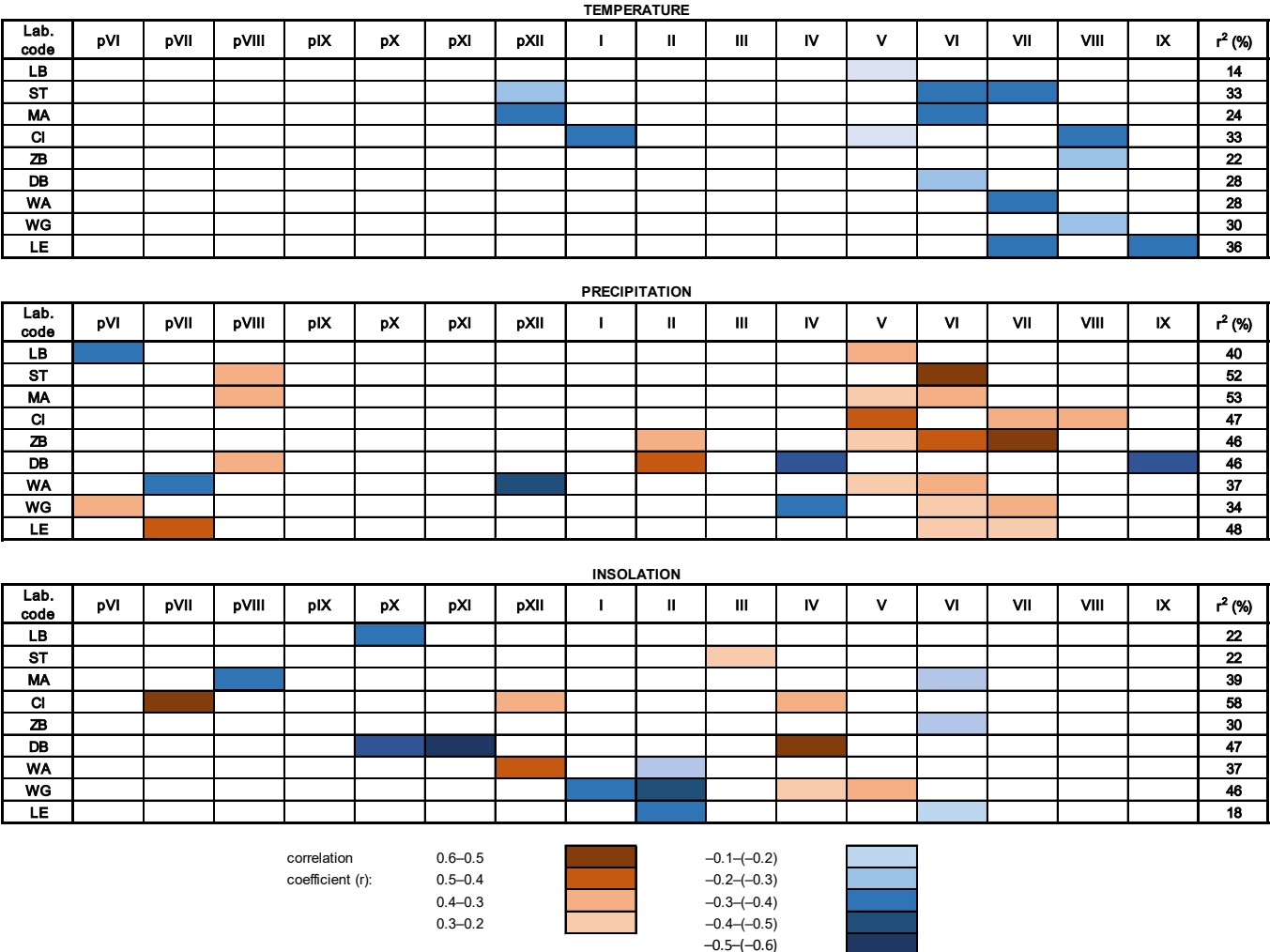

**Figure 3.** Results of correlation analyses (r) for the hawthorn chronologies through the period 1988–2020 (33 years) for temperature, precipitation and insolation. Only statistically significant values ($p \leq 0.05$) are shown; p, previous year; $r^2$, multiple regression coefficient of determination.

### 3.3. Regional Pointer Years

The analysis indicated 10 years, during which in at least 6 local chronologies, over 90% of trees had a lower growth compared to the preceding year (negative pointer years)—1992, 1994, 1998, 2001, 2003, 2008, 2010, 2012, 2015 and 2019—and 4 positive years, characterized by a positive growth trend compared to the preceding year—1999, 2002, 2009 and 2013 (Figures 4 and 5). The analysis of weather conditions in the study area during the designated pointer years enabled us to link the occurrence of negative years predominantly with the occurrence of drought during the summer period. The shortage of rainfall in the spring–summer period (from May to August) was the reason for decreased tree-ring width. Most frequently, negative pointer years are also years with a low annual rainfall sum (considerably lower than average). The average annual air temperature in negative pointer years was most frequently close to average or above average, the temperature of winter and early spring was variable, and summer temperatures were higher than average. The insolation during negative pointer years has a lower significance, as for various years the annual sum of insolation is higher or lower than average. It is frequent, however, for high insolation values to occur in the spring and summer months. The year 1992 may serve as an example of a negative regional pointer year for the hawthorn population in Poland. In

this year, negative growth trends were noted in the following chronologies: LB, ST, MA, CI, ZB, DB and LE (7 out of 9 chronologies). The average annual air temperature for this year was higher than average at all the studied weather stations, the winter was warm, and the summer months were very warm or even hot (e.g., August). The annual rainfall sum was lower than average in all regions (a dry year), and severe rainfall shortages were noted for the period from May or June to the end of summer, especially in August. The annual insolation sum varies depending on the weather station (higher than average for some stations, lower than average for others). Elevated insolation is noted for the summer months, however.

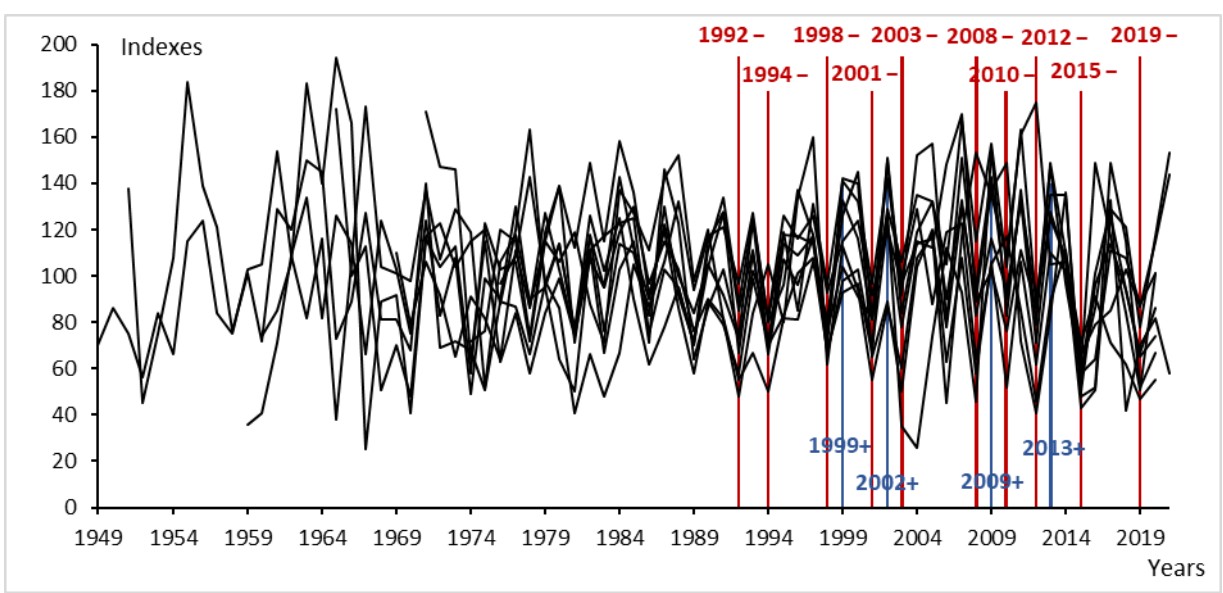

**Figure 4.** Indexed local hawthorn chronologies from Poland and regional pointer years (blue bars and + denote positive years; red bars and—denote negative years).

Air temperature and rainfall in the summer months are the most significant weather elements in the positive pointer years. The average annual temperature during these years is close to the multi-year average or slightly higher, the winter months are cold or warm, and from May to August, the air temperature is often close to average or slightly higher, but with no extreme values. Positive pointer years are years with higher than average annual rainfall sum (humid years). In the summer months, precipitation totals are also higher than average, although single months with rainfall shortages also occur. The insolation during these years (both annual and monthly values) is variable. The year 2013 may serve as an example of a positive regional pointer year in the hawthorn populations in Poland as it occurs in 7 out of 9 chronologies (LB, CI, ZB, DB, WA, WG and LE). The year 2013 was slightly warmer than the multi-year average, the winter was long and cold (negative average temperatures were noted as late as March), and the temperature in the summer months was average or slightly higher than average. It was a humid year (annual rainfall sums are considerably higher than average), but importantly, high rainfall sums are noted also for the summer months, although slight rainfall shortages occurred for single months at several stations. The annual number of hours of sunshine is higher than average or close to average, and in the vegetation season, there are both months with very low and very high insolation values, although high values are noted more frequently.

### 3.4. Dendrochronological Regions

Based on 9 indexed local hawthorn chronologies (for a common period spanning 33 years from 1988 to 2020), we made an attempt to distinguish dendrochronological regions using cluster analysis. The clustering procedure was based on the unweighted pair-group method using arithmetic averages (UPGMA), and the similarity function was

computed using the Pearson correlation method. All computations were performed using Statistica (version 13.3). Based on the analysis of the linkage distance relative to Pearson's 1-r linkage steps, the breakpoint was determined at the height of about 3.5 linkage distance values (Figure 6).

| Years /Plots | LB | ST | MA | CI | ZB | DB | WA | WG | LE |
|---|---|---|---|---|---|---|---|---|---|
| 1988 | | | | | | | | | – |
| 1989 | – | – | – | | – | | | | – |
| 1990 | + | + | + | | | | | | + |
| 1991 | | | | | | | | | |
| **1992** | – | – | – | – | – | – | | | – |
| 1993 | + | + | | | | | + | | + |
| **1994** | – | – | – | – | | – | – | | |
| 1995 | + | | | + | + | | + | | + |
| 1996 | – | | | | | | | | |
| 1997 | + | | | | | | | | |
| **1998** | – | – | – | – | | – | | | – |
| **1999** | | | + | + | + | | + | + | + |
| 2000 | | | | | | | | | |
| **2001** | – | – | – | | – | – | – | – | – |
| **2002** | | + | + | + | | + | | + | + |
| **2003** | – | – | – | – | – | – | – | – | – |
| 2004 | | | | | | + | + | | |
| 2005 | – | | | | | | | | |
| 2006 | | | | | | – | – | | – |
| 2007 | | | | | | + | | | + |
| **2008** | – | | – | – | | – | | – | – |
| **2009** | + | + | | + | + | + | | + | |
| **2010** | | – | – | | – | – | – | – | – |
| 2011 | | + | | - | | | + | | + |
| **2012** | – | – | – | – | | – | | – | – |
| **2013** | + | | | + | + | + | + | + | + |
| 2014 | | | | | | | | | |
| **2015** | – | – | – | – | – | – | – | – | – |
| 2016 | | + | | + | + | | | | |
| 2017 | | | | | | + | | | + |
| 2018 | | | | | | | | | |
| **2019** | – | | – | – | | | – | – | – |
| 2020 | + | | | | | | | | |

**Figure 5.** Summary of pointer years for the local chronologies of the hawthorn in Poland during the period 1988–2020 (33 years). Blue—positive pointer years, orange—negative pointer years, bold years—regional pointer years.

Using a 0.35 linkage distance on the indexed chronology clustering dendrogram, we identified three clusters/regions: I—the western part of the study area; II—the CI study plot; III—the eastern part of the study area (Figure 7). Region I includes the sites: LB, MA, LE, WA, ST and ZB. These populations are located in the northern, central and southern parts of the country, at various elevations (from 45 to 501 m a.s.l.). Their age varies (from 45–47 to 72 years). Also, their average annual growth varies (from 1.42 to 2.41 mm/year). All the sites clustered together in cluster I are located in the western part of the country,

characterized by a milder climate than the eastern part of Poland. Cluster II comprises only one site, CI. The separation of this site from region I is likely caused by habitat or genetic conditions. Region III includes two sites: WG and DB. These tree populations are growing at various elevations (115 and 451 m a.s.l.), are of similar age (50 and 56 years), and are characterized by rather high growth rates (2.34 and 3.25 mm/year). They are also the easternmost study sites.

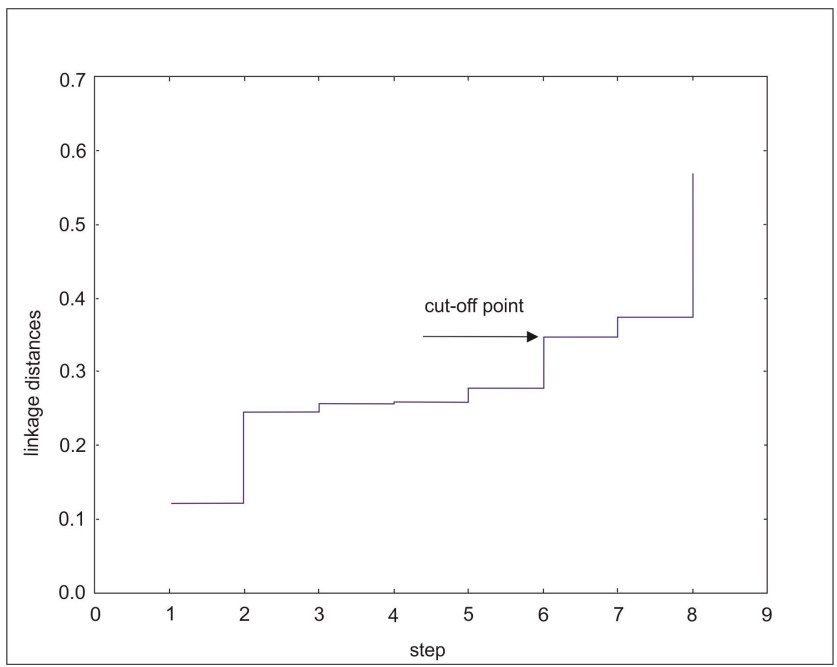

**Figure 6.** Linkage distances (the one minus Pearson's correlation distance) for the local hawthorn chronologies from Poland, through the period 1988–2020.

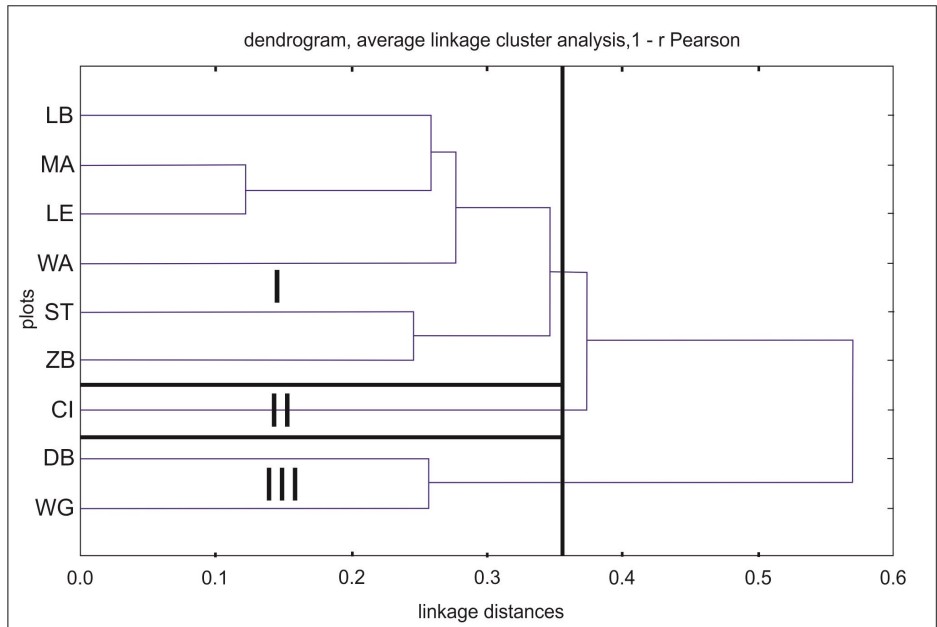

**Figure 7.** Dendrogram of indexed hawthorn chronologies (average linkage cluster analysis, the one-minus Pearson correlation coefficient distance, period covered: 1988–2020). Dendrochronological regions denoted with Roman numerals (I, III) and area (II).

## 4. Discussion

Trees from the genus *Crataegus* are rarely subject to dendrochronological studies. One reason for this could be the difficulty in identifying and precisely dating tree rings. Difficulties while working with this species have been reported by Mills [42] (difficult sample collection)—"a very hard and difficult wood to core"; difficulties in detecting tree rings have been reported by Decuyper et al. [43]; and Cedro and Cedro [27] have reported false rings. Finally, 2–4 rows of cells, immediately adjacent to the tree ring boundary and often pinching out, were commonly observed. Unfortunately, there is a lack of dendrochronological studies (this applies to all hawthorn species), although a few studies include comments on the age of trees, rings and growth rate [27,42–46]. Of the approximately 50 species of hawthorn found in Europe and Asia and 100 species found in North America, the Grissino-Mayer study [47] on the dendrochronological potential of shrubs and trees include only *C. azorolus* L. (codenamed CRAZ), which is assigned a cross-index (CDI) equal to 0, which means that "the species does not crossdate, or no information on crossdating for this species has been published. No or little significance in dendrochronology." *Craraegus monogyna* and *C. xmedia* are also not included in Microscopic Wood Anatomy [48], which only states that the subgenus *Crataegus* does not have heartwood and the anatomical structure of the wood cannot serve as a basis for species identification. Hawthorn wood is diffuse porous, so correctly determining the boundaries of the rings is often problematic. The wood is very hard and yellow-light brown in color.

The form of the hawthorn (shrub or multi-trunk specimens), and its low age, may discourage analyzing tree rings. The hawthorn populations studied in Poland are indeed rather young: the highest number of measured tree rings equals 72 (in the LE population), and most commonly it is less than 50 tree rings (Table 2). However, a specimen of the one-seeded hawthorn (*Crataegus monogyna*) is considered to be the oldest tree in France. It is growing in Aubepines, next to a church. It is thought to have been planted in the 3rd century A.D. and at present it is about 1500 years old [49]. This hawthorn tree is in a very poor condition and clearly dying.

In the studied hawthorn populations, the tree-ring width ranges from 1.42 mm/year (in the WA population) to 3.25 mm/year (in the WG population), giving an average value of 2.05 mm/year for Poland (Table 2). Such low annual growth in the WA population may be caused by strong environmental pollution in the immediate surroundings. Until recently, Wałbrzych was a large coal mining and processing hub, with three large coal mines (the last one closed in the 1990s). A coking plant is still in operation. Considerable reductions in tree-ring widths in areas strongly polluted by the coal-burning industry in the south of Poland are noted also by Barniak and Jureczko [50], who reported numerous missing rings and tree-ring width reductions up to 85% in the 1960s through the 1990s.

Tree-ring widths in *Crataegus azarolus* L. growing at elevations from 1717 to 2280 m a.s.l. in Iran were reported to range from 1.96 to 2.36 mm/year. In this case, however, the focus of the study was on wood properties. The study indicated that these depend on elevation a.s.l., but also on rainfall sums and air temperature [44]. Three individuals of *Crataegus monogyna* examined in Scotland displayed different growth rates: 1.35, 1.38 and 2.17 mm/year [42]. A total of 76 tree rings were measured in each specimen, but due to the difficulties in measuring the rings under bark, and the absence of tree rings adjacent to the core, the trees were dated to around 1909, 1915 and 1926, respectively. It was concluded that the trees represent a remnant of a hedge planted in the first decade of the 20th century, and the contrasting age of the trees resulted from early hedge cutting. The impact of sheep grazing on the development of a hawthorn population was studied in Wales [45]. Unfortunately, this publication does not report tree-ring widths for the specimens examined. The age of the trees is estimated at 10 to 115 years, and mortality is thought to increase substantially beyond 80 years. The growth rate per decade varies and, according to the authors, it is impossible to estimate the age of the hawthorn trees from their girth. Sheep grazing controls the rate of the hawthorn population renewal: the rate of renewal and the number of trees and shrubs decreases with increasing sheep numbers.

Specimens up to 12 m high, and up to 60 years old were reported from New Zealand (South Island), where the hawthorn is considered a noxious and invasive plant [46]. At various sites, the growth rate per year was determined as 1.4, 3.0, 3.3 and 4.8 mm. The highest growth rate was observed where there was no or limited sheep grazing.

The dendroclimatic analyses: correlation and response function analysis and pointer year analysis indicate the May–August period, and air temperature variability and precipitation totals as the main factors responsible for the tree-ring widths in the hawthorn in Poland. Lower than average temperatures through this period and higher precipitation sums cause high cambial activity and the formation of a wide tree ring. Drought, especially in conjunction with heat waves, causes growth depressions. Unfortunately, the lack of dendroclimatic studies on the hawthorn from other regions precludes a comparison of the reaction and growth–climate relationship. The impact of flooding on colonization and development of the hawthorn on flood plains in the Netherlands was studied by Decuyper et al. [43] and Cornelissen et al. [51]. Colonizing by the hawthorn was influenced mostly by the flooding periods. The positive impact was via seed transport and supplying water to plants during dry periods. The negative impact was via flooding seedlings and young plants during extended flooding. Extreme weather events (e.g., droughts or very high or low rainfall) also impacted the hawthorn assemblages on the studied flood plains. Livestock grazing caused the number of hawthorn individuals to decrease in the study plots.

The division into dendrochronological regions in Poland (two regions, I—western and III—eastern, and CI as a single distinguished site) points to a significant role in the degree of climate continentality. The lack of an unambiguous reason for the distinction of the CI site (most likely habitat and genetic conditions) highlights the need for further, detailed study taking into account the habitat and genetic features that differentiate the populations.

## 5. Conclusions

Nine local chronologies for the hawthorn from Poland were compiled, spanning from 45 to 72 years. The average tree-ring width is from 1.42 to 3.25 mm/year. The growth-climate analyses point to the weather conditions of the May–August period as the most important tree-ring shaping factors. Higher than average rainfall and low air temperatures through these months cause the formation of wide tree rings. Conversely, rainfall shortages and heat waves cause growth depressions in the hawthorn. Cluster analysis enabled the identification of two regions, western and eastern, and identified the CI site as a separate cluster. The reason for such clustering is most likely the changes in the degree of climate continentality (the eastern and western regions), and habitat and genetic conditions in the case of the CI site. The obtained results highlight the need for continued studies on this species, compiling a larger number of chronologies from various regions, habitat conditions, from trees as old as possible, and for further dendroclimatic analyses.

**Author Contributions:** A.C. and B.C.: conceptualization, field collection, data analyses, draft preparation, and review and editing. All authors have read and agreed to the published version of the manuscript.

**Funding:** This research received no external funding.

**Data Availability Statement:** Data available in a publicly accessible repository. The data presented in this study are openly available in RepOD at https://doi.org/10.18150/HXG3SO.

**Conflicts of Interest:** The authors declare no conflict of interest.

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
