# Peer review of "Dendrochronological Analysis of One-Seeded and Intermediate Hawthorn Response to Climate in Poland"

_forests, doi:10.3390/f14112264_

Round 1

Reviewer 1 Report

Comments and Suggestions for Authors

The authors took for the study a very labor-intensive and dendrochronologically unstudied tree species – hawthorn (Crataegus).

The value for dendrochronological research of such a short-lived tree species as hawthorn is highly questionable. The chronologies compiled by the authors are only 45-72 years old. Accordingly, the authors were able to analyze only a very short climate series - 33 years (1998-2020), which inevitably affected the results of the study. The conclusion about the significant influence of May-August weather conditions on the radial growth of trees is not new, as is the information about the increase or decrease in tree rings depending on the dynamics of precipitation and temperatures.

The identification of two regions in connection with the continental climate in Poland is of some interest.

I also note that the authors applied classical methods of dendrochronological research, met the necessary requirements on the number of samples selected (at least 20 trees from one site, two samples were taken per tree), carried out the necessary statistical procedures, and analyzed weather data from 16 weather stations.

Despite the above shortcomings, I believe that the article can be published in the journal Forests.

Author Response

MDPI Forests                                                                              Szczecin, 13.11.2023

Dear Editors and Reviewers,

We are grateful for the insightful analysis of our manuscript, and all the comments, and suggestions provided. We did our best to take into account all the remarks.

  We hope that the enclosed revised manuscript meets the requirements of the Editors and Reviewers, and is suitable for publication.

Reviewer 1

All suggestions of Reviewer 1 have been incorporated. We are grateful for the insightful analysis of our manuscript, and all the comments, and suggestions provided. We did our best to take into account all the remarks. We hope that the enclosed revised manuscript meets the requirements of the Editors and Reviewers, and is suitable for publication.

Sincerely,
the authors

Anna Cedro and Bernard Cedro

Reviewer 2 Report

Comments and Suggestions for Authors

The presented data in the manuscript are new and use pre-existing analysis techniques. The objectives of the paper are clear and the quality of the data is appropriate for the analysis. Generally, I consider the paper well-written and easy to follow and have only some minor comments.
The font and size of texts must be the same in all Figures and Tables. The quality of Figures 6 and 7 should be improved. One methodological question: why do the Authors use R2 and not correlation R for the climate-tree growth analysis?

Minor comments:

Abstract

L18-19/L436: "were complied with tree ages between 45 and 72 years"
L26: CI: what is it?

Introduction

L37: claiming: what does it mean?
L43/L55: please change or delete "in the fight"
L59-L72: I would change the order here: put this paragraph to L54 (after "quality and value [18].") and the section L54-58 to L74 (before "This paper..")

MM

L79: missing reference to Fig.1
L88-155: I would shorten the description of the sites preserving only the most relevant information
L157: (Yponomeuta padella): font size
L159: Table1: I think the geographic coordinates are enough to present up to 4 decimal places
L168: "Those trees that were the largest and looked the oldest were selected for sampling" -> "Hawthorn individuals that were dominant and presumably older were selected..."
L170-171: missing: one core per individual
L173: "very weak visibility of borders between the annual growth rings" -> "very weak visibility of annual growth rings"
L176: write in one: "14250"
L189: STD: standardized?
L193-194: "spanning from June of the previous year until September of the current year of ring formation were.."
L196: R2 values? Not correlation (R)?
L206-207: "were retrieved from 16 weather stations of the Institute...:"  

Results

L218: populations
L223-226: repeated content (L221-222)
L226-230: this may be better to mention in the 2.2 section e.g. after L185 and combined with L190-192
L235: Cambial age was determined based on the number of rings at breast height?
L253-254: or "higher temperature does not favor the formation of wide rings"
L255: why determination coefficient? Simple correlation is not sufficient?
L266: April
L284: pluvial? I think it is rather "drought"
L297: rainfall
L324: Is this not a table? The header is missing (years, sites)
L333: reference for statistica?
L336: Figure 6: please modify the y-axis caption to the left side  
L352: missing y-axis caption  

discussion

L358: "in identifying and precisely dating tree rings"
L437-438: this sentence is not necessary

Author Response

Reviewer 2

We are grateful for the insightful analysis of our manuscript, and all the comments, and suggestions provided. We did our best to take into account all the remarks.

  We hope that the enclosed revised manuscript meets the requirements of the Editors and Reviewers, and is suitable for publication.

All suggestions of Reviewer 2 have been incorporated. Specifically, these are:

  1. L18-19/L436 - Corrected according to the reviewer’s suggestion
  2. L26 - CI is plot’s signature
  3. L37 - Corrected according to the reviewer’s suggestion
  4. L43/L55 - Corrected according to the reviewer’s suggestion
  5. L79 - Corrected according to the reviewer’s suggestion
  6. L 88-155 - in the authors' opinion, the description of the sites and, especially, the differences between them is necessary because we refer to it in the discussion part (e.g. separation of dendrochronological regions)
  7. L157 - Corrected according to the reviewer’s suggestion
  8. L159 - Corrected according to the reviewer’s suggestion
  9. L168 - Corrected according to the reviewer’s suggestion
  10. L170-171 - Corrected according to the reviewer’s suggestion (but not one core… but two core per individual)
  11. L173 - Corrected according to the reviewer’s suggestion
  12. L176 - Corrected according to the reviewer’s suggestion
  13. L189 - Corrected according to the reviewer’s suggestion
  14. L193-194 - Corrected according to the reviewer’s suggestion
  15. L 196 - we use correlation values (r) and r2 values (multiple regression determination coefficient), we added an explanation in the caption Figure 3
  16. L206-207 - Corrected according to the reviewer’s suggestion
  17. L218 - Corrected according to the reviewer’s suggestion
  18. L223-226 - these lines contain data on the tree-ring width (Table 2) and the cumulative radial growth (Figure 2).The results are similar.
  19. L226-230 - yes, this sentence fits the methodology, but here the explanation is more clear, according to the authors
  20. L235 - yes, this sentence fits the methodology, but here the explanation is more clear, according to the authors
  21. L253-254 - yes, this sentence fits the methodology, but here the explanation is more clear, according to the authors
  22. L255 - we use correlation values (r) and r2 values (multiple regression determination coefficient), we added an explanation in the caption Figure 3
  23. L266 - Corrected according to the reviewer’s suggestion
  24. L284 - Corrected according to the reviewer’s suggestion
  25. L297 - Corrected according to the reviewer’s suggestion
  26. L324 (Figure 5) - Corrected according to the reviewer’s suggestion
  27. L333 - we added version of Statistica, other authors do not cite this program
  28. L336 (Figure 6) - Corrected according to the reviewer’s suggestion
  29. L352 (Figure 7) - Corrected according to the reviewer’s suggestion
  30. L358 - Corrected according to the reviewer’s suggestion
  31. L437-438 - Corrected according to the reviewer’s suggestion

The language accuracy was re-checked, and corrected where required.

We are grateful for the insightful analysis of our manuscript, and all the comments, and suggestions provided. We did our best to take into account all the remarks. We hope that the enclosed revised manuscript meets the requirements of the Editors and Reviewers, and is suitable for publication.

Sincerely,
the authors

Anna Cedro and Bernard Cedro

Reviewer 3 Report

Comments and Suggestions for Authors

Overview and general recommendation

This article fits well with the aim and scope of the special issue “Tree Growth in Relation to Climate Change” in “Forests”. This article will benefit forestry researchers and especially dendrochronologists who work with broadleave species.

The introduction has explained well the background and value of this specific species. The objective is clearly written. The methodology is well described. The majority of the results were also well written with some minor parts that need to be rephrased better. Even though there were not many dendrochronological studies of hawthorn species, the authors still did a good job linking the local research with a global perspective. The reviewer considers the article needs minor modifications in the methodology and result parts, after that, the paper is ready to publish.

Minor comments

- In general, “monthly total precipitation” is a more common term than “monthly precipitation sum” in dendrochronology research, please consider changing it accordingly.

- In 2.1 Study area: It would be clearer about the climate condition for the 9 experimental plots if there were some temperature, precipitation and insolation chronological figures for the main analysis years.

- In 3.4 Dendrochronological regions: Line 329 to 333 belongs to the methodology. Please consider moving them to the right session.

Comments on the Quality of English Language

- In 3.3 Regional Pointer Years: this part needs to go through grammar checking and rephrasing some of the sentences.

Author Response

MDPI Forests                                                                              Szczecin, 13.11.2023

Dear Editors and Reviewers,

We are grateful for the insightful analysis of our manuscript, and all the comments, and suggestions provided. We did our best to take into account all the remarks.

  We hope that the enclosed revised manuscript meets the requirements of the Editors and Reviewers, and is suitable for publication.

Reviewer 3

All suggestions of Reviewer 3 have been incorporated. Specifically, these are:

- “monthly precipitation sum” was changing to “monthly total precipitation” - Corrected according to the reviewer’s suggestion

- In 2.1 Study area: It would be clearer about the climate condition for the 9 experimental plots if there were some temperature, precipitation and insolation chronological figures for the main analysis years.

authors can prepare climate diagrams, but it will be 9 additional drawings that will take up a lot of space, please decide editor

- In 3.4 Dendrochronological regions: Line 329 to 333 belongs to the methodology. Please consider moving them to the right session

yes, this sentence fits the methodology, but here the explanation is more clear, according to the authors

The language accuracy was re-checked, and corrected where required.

We hope that the Reviewers and Editors find the current form of the article  acceptable for publication in this journal.

Sincerely,
the authors

Anna Cedro and Bernard Cedro